**Data Availability Statement:** All relevant data are within the manuscript and its Supporting Information files.

# Comparison of optimal bowel cleansing effects of 1L polyethylene glycol with ascorbic acid versus sodium picosulfate with magnesium citrate: A randomized controlled study

Jun Lee[1], Seong-Jung Kim[1], Sang-Wook Kim[2], Hyo-Yeop Song[3], Geom Seog Seo[3], Dong-Hyun Kim[4], Dae-Seong Myung[4], Hyun-Soo Kim[4], Young-Eun Joo[4]*, So Yeong Kim[5]

1 Department of Internal Medicine, College of Medicine, Chosun University, Gwangju, Republic of Korea, 2 Department of Internal Medicine, Jeonbuk National University Medical School, Jeonju, Republic of Korea, 3 Department of Internal Medicine and Digestive Disease Research Institute, Wonkwang University School of Medicine, Iksan, Republic of Korea, 4 Department of Internal Medicine, Chonnam National University Medical School, Gwangju, Republic of Korea, 5 Department of Preventive Medicine, College of Medicine, Chosun University, Gwangju, Republic of Korea

* yejoo@chonnam.ac.kr

## Abstract

Various low-volume bowel cleansing formulations that improve compliance have been approved and are being used in clinical practice. This study aimed to compare the effectiveness of 1 L polyethylene glycol (PEG) with ascorbic acid with that of sodium picosulfate (PICO) with magnesium citrate. This was a multicenter, randomized controlled, non-inferiority study. Patients were randomized into a 1 L PEG with ascorbic acid group and a PICO with magnesium citrate group according to the bowel cleansing agent used. Colonoscopy was performed as a single-blind study wherein the endoscopist had no information about any bowel preparation agent. The efficacy of bowel cleansing was assessed using the Harefield Cleansing Scale (HCS), and adverse events, preferences, and satisfaction were evaluated using a patient-reported questionnaire before colonoscopy. A total of 254 participants were randomly assigned to two groups: 115 in the 1 L PEG with ascorbic acid group and 113 in the PICO with magnesium citrate group. Overall bowel cleansing success was not statistically different between the two groups (97.4 vs. 97.3%), confirming that 1 L PEG with ascorbic acid was not inferior to PICO with magnesium citrate (lower confidence limit, -4.15%; p = 1.00). High-quality bowel cleansing was achieved in 87% of the 1 L PEG with ascorbic acid group and 77% of the PICO with magnesium citrate group (Lower confidence limit, 1.29%, p = 0.05). In terms of patient satisfaction, PICO with magnesium citrate was better, but compliance and side effects were similar in both groups. The 1 L PEG with ascorbic acid showed similar efficacy and adverse events as PICO with magnesium citrate. Although 1 L PEG with ascorbic acid is very effective in bowel preparation despite its small volume, it is necessary to increase satisfaction such as taste and feeling.

**Funding:** This study was funded by TAEJOON PHARM Co., Ltd. Seoul, Korea. The funder had no role in the study design, conduct, analysis, or reporting.

**Competing interests:** The authors declare no potential conflicts of interest.

## Introduction

Colorectal cancer has continued to decline in countries with well-established screening programs for colorectal cancer in recent years [1, 2]. Colonoscopy is the most effective screening method to reduce the incidence and mortality of colorectal cancer by detecting cancer at an early stage and removing adenomatous polyps. However, the effect of colonoscopy on the prevention of colorectal cancer differs by country and region, and a large difference in the detection rate of adenomas among examiners has been reported [3]. Therefore, it is recommended to perform high-quality colonoscopy based on various quality control indicators.

Bowel preparation is one of the most important indicators for quality of the colonoscopy. However, inadequate bowel preparation is reported in approximately a quarter of all colonoscopies, which may reduce the detection rate of adenomas, delay procedure time, and increase the risk of side effects of the procedure [4, 5]. Although the 4 L polyethylene glycol (PEG) split-dose regimen provides high quality bowel cleansing, its large volume results in poor satisfaction, tolerability, and compliance [6, 7]. Various low-dose (2 L) cleansing agents have been newly developed, and several studies have confirmed that they are not inferior to 4 L PEG [8, 9]. However, low-dose cleansing agents still have a large volume and bad taste, so there is no preparation that can satisfy all patients so far [10]. Recently, a 1 L PEG formulation that increased the ascorbic acid content and reduced the volume by half from the existing 2 L PEG was released. Some studies have reported that it is not inferior to 2 L PEG with ascorbic acid, confirming similar results in terms of adequate preparation, side effects, and preference [11–13]. However, studies comparing 1 L PEG with ascorbic acid with other non-PEG based low-volume cleansing agents are insufficient to date [14–16]. In particular, there are only two studies comparing sodium picosulfate (PICO) with magnesium citrate. However, in one study, the data were unreliable due to major errors in study design [14]. In other study, since the Aronnchick scale was applied as a bowel preparation index, there is a limitation in that it is difficult to confirm whether appropriate efforts to remove residual debris have been implemented [16]. The randomized controlled studies of 1 L PEG with ascorbic acid are summarized in Table 1.

Therefore, this study evaluated the efficacy, safety, compliance, and satisfaction of bowel preparation formulation by comparing the efficacy of 1 L PEG with ascorbic acid with that of PICO with magnesium citrate.

## Methods

### Study design and patients

We conducted a multicenter, prospective, single-blinded, randomized, controlled study at four tertiary academic hospitals. From July 2019 to June 2021, outpatients aged between 19 and 75 years who visited the hospital to undergo colonoscopy were enrolled. Patient registration was conducted through competitive registration at four tertiary academic hospitals. Patients with a history of gastrointestinal tract surgery, inflammatory bowel disease, intestinal obstruction or pseudo-obstruction, impaired renal function, pregnancy, recent malignancy, severe constipation, taking laxatives or gastrointestinal motility drugs within the past three months, or patients who did not consent to the study were excluded. Eligible patients who written informed consent were randomly assigned to the 1 L PEG with ascorbic acid group and PICO with magnesium citrate group in a 1:1 ratio using computer-generated randomization. The patients answered the patient-reported outcome questionnaire about the adverse events, preference, and satisfaction with the cleansing agents before they underwent colonoscopy. (S1 File) Colonoscopy was performed as a single-blind study in which the endoscopist had no

**Table 1. Summary data on randomized controlled trial for 1L polyethylene glycol with ascorbic acid.**

| Study | Year | Efficacy (bowel cleansing success rate) | Safety | Compliance & tolerability | Comment & Limitation |
|---|---|---|---|---|---|
| 1L PEG with ascorbic acid vs 2L PEG | | | | | |
| Hong et al. [11] | 2021 | Non-inferior to 2L PEG (93.1% vs 91.8%, p<0.661) | Higher adverse events than 2L PEG (65.7% vs 52.9%, P = 0.015) | Similar to 2L PEG | |
| Koo et al. [12] | 2021 | Non-inferior to 2L PEG (99.0% vs 94.9%, p = 0.100) | Similar to 2L PEG (16.5% vs 8.2%, p = 0.287) | Similar to 2L PEG | |
| Bisschops et al. [13] | 2019 | Superior to 2L PEG (97.3% vs 92.2%, p = 0.014) | Similar to 2L PEG (11.5% vs 7.6%, p = 0.140) | Similar to 2L PEG | |
| 1L PEG with ascorbic acid vs non-PEG based low-volume cleansing agent | | | | | |
| DeMicco et al. [15] | 2018 | Non-inferior to Trisulfate (85.1% vs 85.0%, p = 0.528) | Similar to Trisulfate (14.9% vs 9.4%, p = N-A*) | Similar to Trisulfate | • Relatively too low overall success rate in both groups<br>• different diet control |
| Schreiber et al. [14] | 2019 | Non-inferior to Sodium picosulfate with magnesium citrate (62.0% vs 53.8%, p = 0.04) | Higher adverse events than Sodium Picosulfate with magnesium citrate (17.0% vs 10.0%, p = 0.03) | Similar to Sodium picosulfate with magnesium citrate | • Relatively too low overall success rate in both groups<br>• no applicable split regimen (different administration time) |
| Kojecky et al. [16] | 2019 | Non-inferior to Sodium picosulfate with magnesium citrate (86.2% vs 72.5%, p = 0.023) | Higher to Sodium Picosulfate with magnesium citrate (34.0% vs 10.4%, p<0.009) | Lower tolerance to Sodium picosulfate with magnesium citrate (55.9 vs 82.3%, p<0.00.) | • Relatively too low overall success rate in both groups<br>• Using 5 points scale for bowel cleansing scale |
| Our study | 2022 | Non-inferior to Sodium picosulfate with magnesium citrate (97.4% vs 97.3%, p = 1.00) | Similar to Sodium picosulfate with magnesium citrate | Similar to Sodium picosulfate with magnesium citrate | Better high-qualtiy bowel cleansing rate than Sodium picosulfate with magnesium citrate (87% vs 77%, p = 0.05) |

N-A

*: not available data. PEG: polyethylene glycol

information about any bowel cleansing agent. The study protocol was approved by the Institutional Review Board Committee of the Chosun University Hospital (2019-05-0008) and performed in accordance with Declaration of Helsinki and Good Clinical Practice guidelines and local regulations. The trial was registered in the Korean Clinical Trial Registry (KCT 0004595; https://cris.nih.go.kr). Although it was registered with the KCT registry 3 months after the start of the multicenter study due to a paperwork error, the authors confirm that all relevant ongoing trials for this intervention have been registered.

## Bowel preparation

Verbal and written instructions for bowel preparation were given to all participants in the 1 L PEG with ascorbic acid group and PICO with magnesium citrate group through a common color printed leaflet on diet control and drug administration from a research nurse at each hospital. All participants were instructed to eat a low-residue diet three days before colonoscopy and a clear liquid diet before colonoscopy. Subsequently, only a small amount of clear water was allowed to prevent dehydration until 3 hour(h) before colonoscopy. Both groups were administered a bowel cleansing agent using a split-dose regimen. In the 1 L PEG with ascorbic acid group, the participants ingested 500 mL of PEG solution (CleanViewAL; Taejoon Pharm. Inc., Seoul, Korea) for 30 min starting at 9 p.m. on the day before the procedure, followed by an additional 500 mL of water. They ingested another 500 mL of PEG solution with 500 mL water in the same way 4 h before colonoscopy. In the PICO with magnesium citrate group, the participants ingested one bottle of 170 mL PICO with magnesium citrate (Picosolution; Pharambio Korea Co., Ltd., Seoul, Korea) with 1 L of water for 2 h from 9 pm on the day

before the procedure. They ingested another bottle of 170 mL PICO with magnesium citrate in the same way 5 h before colonoscopy. The clinical research nurse evaluated adherence to medication prior to colonoscopy, and the endoscopists performing the examination did not know which medication the participant was taking.

## Outcome assessments

The primary endpoint was the success rate of bowel preparation between the two groups. Bowel cleansing success is defined as an A or B on the Harefield cleansing scale (HCS), which corresponds to a score of 2 or higher in all segments, whereas bowel cleansing failure is defined as a C or D on the HCS, where at least one segment scored 0 or 1 [17]. The secondary endpoints were the rate of high-quality bowel cleansing, patient satisfaction, compliance, and safety. High-quality bowel cleansing is defined by the A of the HCS, in which all segments score three or four. Patient tolerability and adverse events were evaluated using a patient outcome questionnaire prepared prior to colonoscopy. Taste, feel, ease of taking, and overall satisfaction were evaluated on a scale of 1–10 using the visual analog scale (from 10 = most satisfactory to 1 = most dissatisfied). Medication compliance and whether or not to take the same drug again the next time were also evaluated. Medication compliance was defined as the case in which each bowel cleansing agent was completely administered according to the bowel preparation method. Regarding adverse events, all possible complications, such as nausea, vomiting, abdominal pain, abdominal distension, insomnia, and paresthesia, were checked.

## Statistical analysis

The appropriate sample size was calculated based on a previous phase 3 trial [14]. The ratio between the 1 L PEG with ascorbic acid group and the PICO with magnesium citrate group was assumed to be 1:1, and appropriate with target of the test drug and the comparator drug was P1 (1 L PEG) = 0.83, P2 (PICO) = 0.95, significance level α = 0.05, power 80% [13, 18]. The HCS of the 1 L PEG with ascorbic acid group was compared with that of the PICO with magnesium citrate group using a non-inferiority test. Non-inferiority was confirmed only if the one-sided 97.5% lower confidence limit for the difference between treatments was ≥-10. The minimum number of patients required for the clinical trial was 117 for each group. Considering a dropout rate of approximately 10% in each group, there were 127 subjects in each group, for a total of 254 subjects.

To determine the characteristic differences between the 1L PEG group and the PICO group, continuous variables are reported as mean (standard deviation, SD), and discrete data are expressed as numbers and percentages. Categorical data were analyzed using frequency (%) analyses, the $\chi 2$ test, and Fisher's exact test, and quantitative data were analyzed using the independent sample t-test. Statistical significance was defined as $p < 0.05$. All statistical analyses were performed using SPSS Statistics version 26.0 (IBM Corp., Armonk, NY, USA).

## Results

### Patient's basal characteristics

From July 2019 to June 2021, a total of 254 patients from four tertiary, academic hospitals participated in the study; 22 patients withdrew their consent before taking the study agent for personal reasons and concerns regarding the risk of the coronavirus disease 2019 (COVID-19). Another four patients were excluded because two patients refused colonoscopy due to nausea and vomiting while taking the drug, one patient took another laxative at the same time, and one patient did not take it according to the protocol. With a total of 115 patients in the 1 L

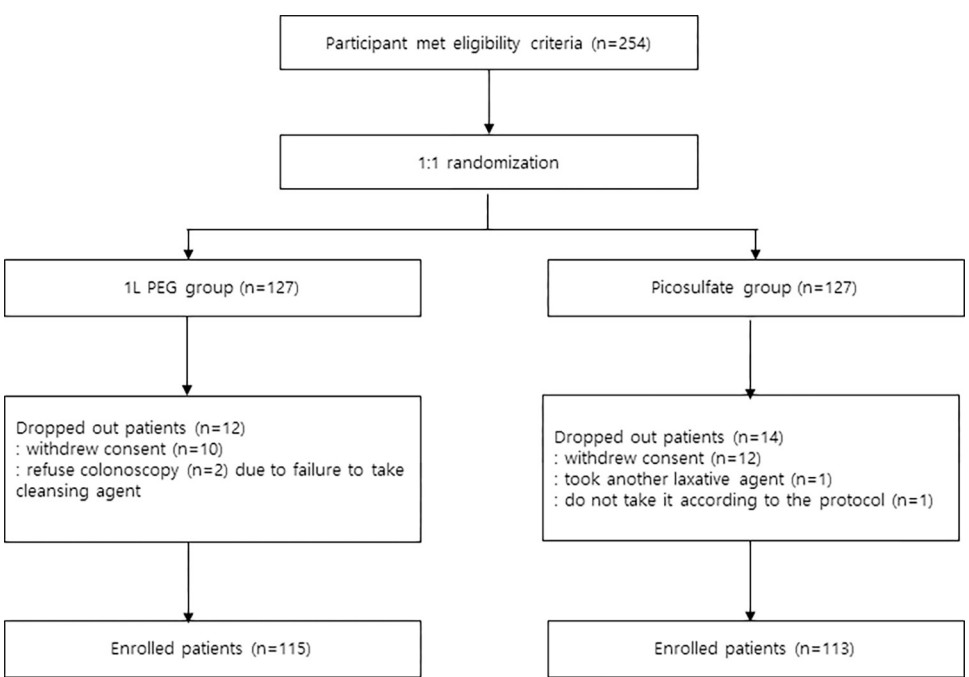

**Fig 1. Flow diagram of the study.** 1L PEG group, 1L polyethylene glycol with ascorbic acid group. Picosulfate group, sodium picosulfate with magnesium citrate.

PEG with ascorbic acid group and 113 patients in the PICO with magnesium citrate group, a total of 228 patients were enrolled. (Fig 1).

The mean age of the patients in the 1 L PEG with ascorbic acid group and PICO with magnesium citrate group was 53.8±12.4 years and 53.7±10.6 years (p = 0.965). There were also no statistically significant differences in sex, body mass index, or comorbidity between the two groups. In both groups, the number of patients who had never used a bowel cleansing agent before was similar (44.3% vs. 44.2%, p = 0.809). Colonoscopy was mostly performed for asymptomatic screening and surveillance in both groups (113 vs. 111 patients, 98.3 vs. 98.2%, p = 0.888). There was no statistically significant difference in adenoma detection between the two groups (39.1% vs. 50.4%, p = 0.086). The overall baseline characteristics of the 1 L PEG with ascorbic acid group and PICO with magnesium citrate group are shown in Table 2.

## Efficacy of bowel cleansing

Overall bowel cleansing success was achieved in 97.4% (112/115) of the 1 L PEG with ascorbic acid group and 97.3% (110/113) of the PICO with magnesium citrate group (lower confidence limit (LCL); -4.15%, p = 1.00). Based on these results, the lower limit of the 97.5% one-sided confidence interval for the 0.1% difference in effectiveness between the two groups, -4.15%, was greater than the non-inferiority limit of -10%; therefore, the 1 L PEG with ascorbic acid group was not inferior to the PICO with magnesium citrate group (Fig 2). In addition, there was no difference between the two groups in bowel cleansing success (segment score ≥2) for each segment. High-quality bowel cleansing was achieved in 87% (100/115) of patients in the 1 L PEG with ascorbic acid group and 77% (87/113) of patients in the PICO with magnesium citrate group (LCL; 1.29%, p = 0.05). The lower limit of 1.29% of the 97.5% one-sided confidence interval for the 10% difference in effectiveness between the two groups was greater than the

**Table 2. Baseline characteristics of the patients according to bowel cleansing agent.**

| | 1L PEG with ascorbic acid group (N = 115) | PICO with magnesium citrate group (N = 113) | p-value |
|---|---|---|---|
| Mean Age (years) | 53.77±12.35 | 53.70±10.64 | 0.965 |
| Sex, Male (n, %) | 54(47.0) | 55(48.7) | 0.795 |
| BMI (kg/m$^2$) | 24.21±3.10 | 23.95±3.63 | 0.572 |
| Comorbidities | | | |
| Hyperlipidemia (n, %) | 12(10.4) | 5(4.4) | 0.084 |
| Diabetes mellitus (n, %) | 9(7.8) | 11(9.7) | 0.611 |
| Hypertension (n, %) | 15(13.0) | 10(8.8) | 0.311 |
| Others (n, %) | 19(16.5) | 27(23.9) | 0.165 |
| Previous history of taking bowel cleansing agent (n, %) | | | 0.809 |
| No | 51(44.3) | 50(44.2) | |
| 2L PEG | 16(13.9) | 19(16.8) | |
| 4L PEG | 48(41.7) | 44(38.9) | |
| Indication of colonoscopy (n, %) | | | 1.000 |
| Screening | 51(44.3) | 50(44.2) | |
| Surveillance | 62(53.9) | 61(54.0) | |
| Diagnostic | 2(1.7) | 2(1.8) | |
| Findings of colonoscopy | | | |
| Adenoma (n, %) | 45(39.1) | 57(50.4) | 0.086 |
| Advanced adenoma (n, %) | 0(0.0) | 0(0.0) | |

PEG: polyethylene glycol. PICO: sodium picosulfate. BMI: body mass index

non-inferiority limit of -10%, the test group was not inferior to the control group. Therefore, even in high-quality bowel cleansing, the 1 L PEG with ascorbic acid group was not inferior to the PICO with magnesium citrate group (Fig 3).

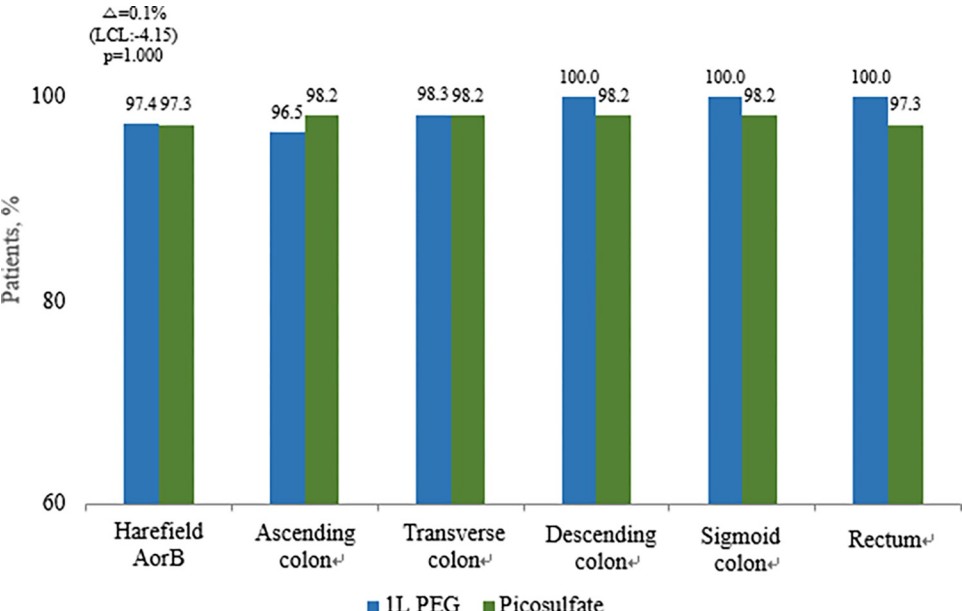

**Fig 2. Efficacy of bowel cleansing using the Harefield Cleansing Scale, including overall and segment bowel cleansing success rate.** Δ–Difference in efficiency between the 1 L PEG with ascorbic acid group and the PICO with magnesium citrate group; LCL- 97.5% one-sided lower confidence limit for the difference between treatments.

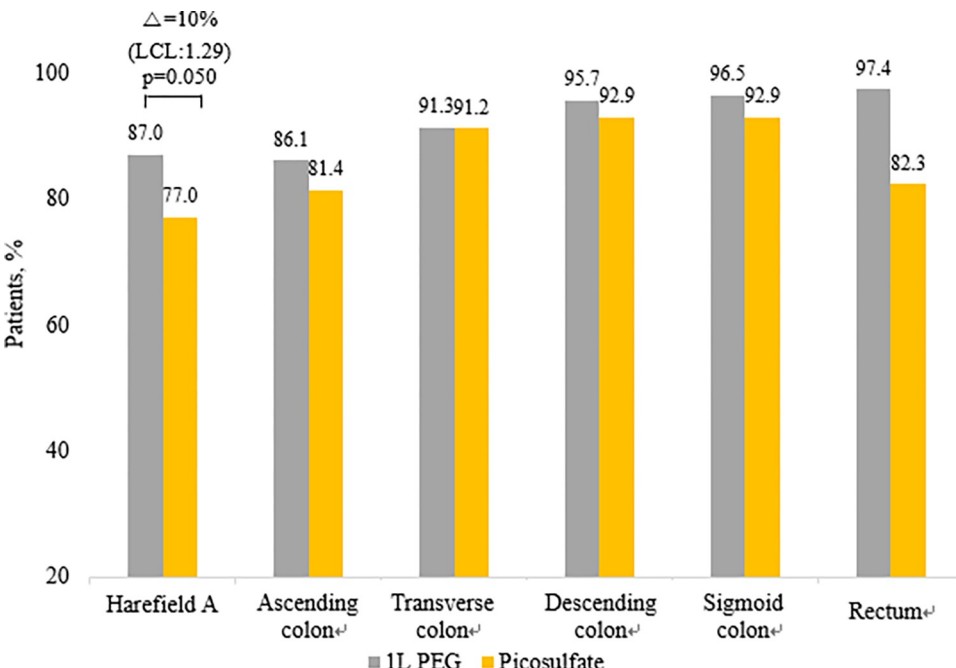

**Fig 3. Efficacy of bowel cleansing using the Harefield Cleansing Scale, including overall and segment high-quality bowel cleansing rate.** Δ–Difference in efficiency between the 1 L PEG with ascorbic acid group and the PICO with magnesium citrate group; LCL- 97.5% one-sided lower confidence limit for the difference between treatments.

## Patient satisfaction and acceptability

PICO with magnesium citrate was preferred over 1 L PEG with ascorbic acid in terms of taste (7.74±1.73 vs. 7.05±2.17), feeling (7.73±1.71 vs. 7.03±2.02), and ease of drug administration (8.35±1.52 vs. 7.71±1.91), and overall satisfaction (8.12±1.38 vs. 7.60±1.96) was also statistically significantly higher (Table 3). However, there was no difference between the two groups in terms of compliance with the bowel cleansing agent (98.3% vs. 99.1%, p = 0.571), and there was no significant difference in terms of accepting the same drug prescription the next time (87.8% vs. 94.7%, p = 0.067).

## Adverse events

Nausea (26.1% vs. 21.2%, p = 0.389) and abdominal pain (12.2% vs. 5.3%, p = 0.067) were the most common adverse events, and the 1 L PEG with ascorbic acid group had a high incidence,

**Table 3. Satisfaction and acceptability of the bowel cleansing agents.**

| | 1L PEG with ascorbic acid group (N = 115) | PICO with magnesium citrate group (N = 113) | p-value |
|---|---|---|---|
| Satisfaction | | | |
| Taste | 7.05±2.17 | 7.74±1.73 | 0.008 |
| Feeling | 7.03±2.02 | 7.73±1.71 | 0.006 |
| Easy of taking agent | 7.71±1.91 | 8.35±1.52 | 0.006 |
| Overall satisfaction | 7.60±1.96 | 8.12±1.38 | 0.023 |
| Medication compliance * | 113(98.3) | 112(99.1) | 0.571 |
| Acceptance to take the same agent for next colonoscopy | 101(87.8) | 107(94.7) | 0.067 |

Medication compliance was defined as the case in which the bowel cleansing agent was completely administered according to the bowel preparation method.

PEG: polyethylene glycol. PICO: sodium picosulfate

**Table 4. Adverse events associated with bowel cleansing agents.**

| | 1L PEG with ascorbic acid group (N = 115) | PICO with magnesium citrate group (N = 113) | p-value |
|---|---|---|---|
| Nausea | 30(26.1) | 24(21.2) | 0.389 |
| Vomiting | 3(2.6) | 5(4.4) | 0.351 |
| Thirstiness | 0(0.0) | 1(0.9) | 0.496 |
| Abdominal pain | 14(12.2) | 6(5.3) | 0.067 |
| Abdominal distension | 1(0.9) | 1(0.9) | 0.747 |
| Fecal incontinence | 0(-) | 0(-) | |
| General weakness | 4(3.5) | 2(1.8) | 0.346 |
| Tingling sensation | 3(2.6) | 2(1.8) | 0.508 |
| Insomnia | 6(5.2) | 1(0.9) | 0.062 |
| Other neurologic adverse events.: mental change/ seizure/ dizziness | 0(-) | 0(-) | - |

PEG: polyethylene glycol. PICO: sodium picosulfate

but there was no statistically significant difference (Table 4). Gastrointestinal side effects such as vomiting, thirstiness, abdominal distension, and fecal incontinence were rare, and there was no difference between the two groups. Systemic side effects such as general weakness and neurological symptoms, including insomnia and tingling sensation, are very rare.

## Discussion

In this study, we confirmed an overall high rate of adequate bowel cleansing (97.4%, 112/115) and compliance (98.3%, 113/115) with 1 L PEG with ascorbic acid. Although 2 L PEG has noticeably improved dosage, many people are still reluctant to undergo colonoscopy because of the large amount of bowel cleansing agent [19, 20]. Recently, some studies have been conducted to reduce the amount of PEG to 1 L to improve patient satisfaction and compliance. Two studies reported that 1 L PEG with oral or suppository bisacodyl had similar effects and reduced patient discomfort compared to 2 L PEG [21, 22]. However, in most guidelines, adjunctive agents such as bisacodyl before colonoscopy do not show any consistent effect and are not routinely recommended [6, 7]. Recently, 1 L PEG with high ascorbic acid content and reduced volume by half has been released and has been approved by the European and Korean Food and Drug Administration (FDA). In the phase 3 study in Korea, 1 L PEG with ascorbic acid was not inferior to the bowel preparation of 2 L PEG (99.0 vs. 94.9%, p = 0.733), and it was confirmed that it was rather superior high quality bowel cleansing in the right colon in the subgroup analysis (83.2 vs. 62.5%, p = 0.005) [12].

Comparative studies with 1 L PEG and other low-dose cleansing agents such as PICO with magnesium citrate are very rare. Schreiber et al. reported that 1 L PEG was not inferior to PICO with magnesium citrate in a phase 3 study [14]. However, this study did not follow the split-dose bowel cleansing regimen. Bowel cleansing agents were administered the day before the colonoscopy, and the administration time was also different between them. Therefore, to the best of our knowledge, our study is the first to confirm that 1 L PEG with a standard split-dose regimen is not inferior to a non-PEG-based low-volume cleansing agent.

High-quality bowel preparation, which is more rigorous than adequate bowel preparation, has recently been emphasized as it contributes to improved adenoma detection rates [23]. Our study confirmed that 1 L PEG was superior to high-quality bowel preparation in the entire colon.(87% vs. 77%, p = 0.05) and not inferior, even when analyzed for each segment.

However, there was no difference in the adenoma detection rate between the two groups. This may be because the sample size was not large, and the screening rate was less than 50% for indications of colonoscopy.

According to recent studies, the frequency of adequate bowel preparation is very high, mostly around 95%, and there are no significant differences between the various bowel cleansing agents. Therefore, patient satisfaction and compliance have become important factors to consider when choosing a bowel cleansing agent. When compared to other low-volume bowel cleansing agents, 2 L PEG-ascorbic acid had low tolerability and satisfaction [10, 24]. In our study, similar to previous studies, overall patient satisfaction, including taste, feel, and ease of dosing, was superior to PICO with magnesium citrate than to 1 L PEG with ascorbic acid. However, it should be noted that there was no difference between the two groups in terms of medication compliance and acceptability of taking the same agent for the next colonoscopy. A relatively small volume of 1 L is considered to improve compliance and acceptability. To increase patient satisfaction, additional research is needed to improve the taste and flavor of bowel cleansing agents in a small volume of 1 L.

There was no difference in gastrointestinal side effects, such as nausea, vomiting, and abdominal pain, and neurological side effects, such as insomnia and tingle sensation. High concentrations of ascorbic acid are known to have the potential to cause nausea and vomiting. In a study comparing 1 L PEG with ascorbic acid and 2 L PEG with ascorbic acid, a higher incidence of vomiting was reported with 1 L PEG with ascorbic acid [13]. However, it was only higher in 1 L PEG with ascorbic acid taking the same day regimen (6.3% vs. 1.1%, p = 0.002). In our study, all patients received a split-dose regimen, and there was no difference in the incidence of vomiting. Taking the 1 L PEG formulation as a split dose and consuming as much clear water as possible is thought to reduce the occurrence of symptoms. Further studies with larger sample sizes are warranted.

This study had some limitations. First, the proportion of patients (10.2%, 26/254) who dropped out in this study was higher than that reported in previous studies. We confirmed the reason for dropping out of patients via phone, and most (84.6%, 22/26) were patients who refused to visit the hospital due to the spread of COVID-19. Therefore, the effect of a high drop rate on the results of our study was considered low. Second, in our study, laboratory tests could not be performed as a side effect of the bowel cleansing agent, and only the patient-reported outcome questionnaire was confirmed. However, in other phase 3 clinical trials, the incidence of electrolyte abnormalities was very low. According to the study protocol, patients with renal impairment, the elderly, and patients taking other laxatives were excluded from the study enrollment; therefore, the possibility of electrolyte imbalance is thought to be very low.

Despite these limitations, our study is a well-designed, multicenter, randomized controlled study comparing 1 L PEG with ascorbic acid and PICO with magnesium citrate. We provided important information in two respects, compared to many other recent bowel cleansing agent studies. First, 1 L PEG with ascorbic acid was not inferior to 2 L PICO with magnesium citrate in terms of efficacy and was rather effective for high-quality cleansing. Second, although 1 L PEG with ascorbic acid was inferior to PICO with magnesium citrate in terms of satisfaction, there was no difference in compliance and acceptance of the same drug.

In conclusion, optimal bowel preparation is an important quality factor in colonoscopy that can improve the efficiency of examination and increase the detection of adenomas. 1 L PEG with ascorbic acid was not inferior to PICO with magnesium citrate in adequate bowel preparation in the general population and was rather effective in overall high-quality bowel preparation. Further studies are needed to prove its effectiveness in patients with risk factors for poor bowel preparation, such as constipation, and safety in high-risk groups, such as patients with inflammatory bowel disease. Importantly, 1 L PEG with ascorbic acid is a bowel cleansing

agent with confirmed efficacy and safety, and will contribute to broadening the options for bowel preparation.

## Supporting information

**S1 Checklist. CONSORT 2010 checklist of information to include when reporting a randomised trial**\*.
(DOC)

**S1 File. Patient reported outcome questionnaire.**
(PDF)

**S2 File.**
(ZIP)

## Acknowledgments

The authors thank Su Young Kim and Seung Woo Lee.

## Author Contributions

**Conceptualization:** Jun Lee, Sang-Wook Kim, Geom Seog Seo, Hyun-Soo Kim, Young-Eun Joo.

**Data curation:** Jun Lee, Seong-Jung Kim, Sang-Wook Kim, Hyo-Yeop Song, Geom Seog Seo, Dong-Hyun Kim, Dae-Seong Myung, Hyun-Soo Kim, Young-Eun Joo.

**Formal analysis:** Jun Lee, So Yeong Kim.

**Funding acquisition:** Jun Lee, Young-Eun Joo.

**Investigation:** Jun Lee, Sang-Wook Kim, Geom Seog Seo, Hyun-Soo Kim, Young-Eun Joo.

**Methodology:** Jun Lee, Sang-Wook Kim, Geom Seog Seo, Hyun-Soo Kim, Young-Eun Joo, So Yeong Kim.

**Writing – original draft:** Jun Lee, Young-Eun Joo.

**Writing – review & editing:** Jun Lee, Sang-Wook Kim, Geom Seog Seo, Hyun-Soo Kim, Young-Eun Joo.

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
