## [Decision Letter · Decision Letter 0]

12 Sep 2022

PONE-D-22-01108Comparison of Optimal Bowel Cleansing Effects of 1L Polyethylene Glycol with Ascorbic Acid versus Sodium Picosulfate with Magnesium Citrate: A Randomized Controlled StudyPLOS ONE

Dear Dr. Joo,

Thank you for submitting your manuscript to PLOS ONE. After careful consideration, we feel that it has merit but does not fully meet PLOS ONE’s publication criteria as it currently stands. Therefore, we invite you to submit a revised version of the manuscript that addresses the points raised during the review process.

Your manuscript was evaluated by two reviewers, their comments are below.

One reviewer raised a concern about how this work contributes to the field in light of work published previously on bowel cleansing regimens, including low dose and split dose. Although you discuss the prior literature in the Discussion, please expand your Introduction to discuss clearly what was previously published in this area and then to provide the rationale for your study in light of the previous findings. It would be helpful to have a table within the Introduction that summarizes previous relevant trials, the regimens examined, and the results with regard to efficacy and tolerability (patient preferences). There are at least two highly relevant studies which were not cited or discussed in your manuscript: see doi: 10.18999/nagjms.83.4.787 and PMID: 31398987.

The second reviewer made several points that need to be addressed with regard to the study's reporting and analyses. As requested by the reviewer, please add a section to your Methods section devoted to statistical analyses, and within this subsection provide information about your analyses in sufficient detail to enable others to replicate your work.

Please address all of the reviewers' comments through revisions to your manuscript and in a Response to Reviewers document.

We look forward to receiving your revised manuscript.

Kind regards,

Renee Hoch, Ph.D.

Managing Editor, PLOS Publication Ethics

PLOS ONE

Journal Requirements:

2. Thank you for submitting your clinical trial to PLOS ONE and for providing the name of the registry and the registration number. The information in the registry entry suggests that your trial was registered after patient recruitment began. PLOS ONE strongly encourages authors to register all trials before recruiting the first participant in a study.

a) your reasons for your delay in registering this study (after enrolment of participants started);

b) confirmation that all related trials are registered by stating: “The authors confirm that all ongoing and related trials for this drug/intervention are registered”.

3. Please provide additional details regarding participant consent. In the ethics statement in the Methods and online submission information, please ensure that you have specified whether consent was written or verbal/oral. If consent was verbal/oral, please specify:

a) whether the ethics committee approved the verbal/oral consent procedure,

b) why written consent could not be obtained, and

c) how verbal/oral consent was recorded. If your study included minors, please state whether you obtained consent from parents or guardians in these cases. If the need for consent was waived by the ethics committee, please include this information.”

“This study was funded by TAEJOON PHARM Co., Ltd. Seoul, Korea. The funder had no role in the study design, conduct, analysis, or reporting.”

“This study was funded by TAEJOON PHARM Co., Ltd. Seoul, Korea. The funder had no role in the study design, conduct, analysis, or reporting.”

Reviewers' comments:

Reviewer's Responses to Questions

**Comments to the Author**

1. Is the manuscript technically sound, and do the data support the conclusions?

Reviewer #1: Partly

Reviewer #2: No

2. Has the statistical analysis been performed appropriately and rigorously? 

Reviewer #1: Yes

Reviewer #2: No

3. Have the authors made all data underlying the findings in their manuscript fully available?

Reviewer #1: Yes

Reviewer #2: Yes

4. Is the manuscript presented in an intelligible fashion and written in standard English?

Reviewer #1: Yes

Reviewer #2: Yes

5. Review Comments to the Author

Reviewer #1: This paper represents a well-written scientific work. The English language is correct and appropriate. Various other investigations were published in this yield, so the subject can stimulate scientific discussion poorly. For this reason, I 'm not fully convinced to support the progression of publication process.

Reviewer #2: A two-arm randomized controlled study was conducted which aimed to compare the effectiveness of PEG to PICO for colon cleansing. The rates of overall bowel cleansing were similar in the two arms.

Major revisions:

1- Include a comprehensive statistical analysis section which lists and describes all the statistical testing methods used to generate the results.

2- Line 138: State the statistical testing method which archives 80% power. Perhaps it was a chi-square test. Thus the trial was designed as a superiority trial, instead of a non-inferiority trial as noted in the abstract. Rectify these differences. Either the trial is a superiority study or a non-inferiority study, not both.

Minor revisions:

1- The standard statistical term for average is mean.

2- Line 167: Provide standard deviations corresponding to the mean ages in the two arms.

3- Line 181: More clearly describe the LCL. It appears to be the lower confidence limit for the difference in bowel cleansing rates.

4- Explain all acronyms and abbreviations at first mention.

6. PLOS authors have the option to publish the peer review history of their article (what does this mean?). If published, this will include your full peer review and any attached files.

Reviewer #1: **Yes: **Giovanni

Cestaro

Reviewer #2: No

---

## [Author Response · Author response to Decision Letter 0]

11 Oct 2022

Editor’s comment

One reviewer raised a concern about how this work contributes to the field in light of work published previously on bowel cleansing regimens, including low dose and split dose. Although you discuss the prior literature in the Discussion, please expand your Introduction to discuss clearly what was previously published in this area and then to provide the rationale for your study in light of the previous findings. It would be helpful to have a table within the Introduction that summarizes previous relevant trials, the regimens examined, and the results with regard to efficacy and tolerability (patient preferences). There are at least two highly relevant studies which were not cited or discussed in your manuscript: see doi: 10.18999/nagjms.83.4.787 and PMID: 31398987

Author’s response: 

As pointed out by the reviewer, we have sincerely revised it. please see the response to reviewers. We've inserted tables and related journals. However, we excluded journals published after our manuscript submission. And among the journals recommended by editor doi: 10.18999/nagjms.83.4.787 "Low dose polyethylene glycol divided doses are not low, but they are less preferred than same-day intestinal preparations for afternoon colonoscopy." is excluded. Because it was a study in which 15ml of sodium picosulfate with magnesium citrate was additionally administered.

The second reviewer made several points that need to be addressed with regard to the study's reporting and analyses. As requested by the reviewer, please add a section to your Methods section devoted to statistical analyses, and within this subsection provide information about your analyses in sufficient detail to enable others to replicate your work.

Author’s response:

We thoroughly consulted with statisticians and came up with an appropriate answer. . please see the response to reviewers

Additional requirements by the editor. 

Author’s response: 

We revised to our manuscript meets PLOS ONE’s style. 

2. Thank you for submitting your clinical trial to PLOS ONE and for providing the name of the registry and the registration number. The information in the registry entry suggests that your trial was registered after patient recruitment began. PLOS ONE strongly encourages authors to register all trials before recruiting the first participant in a study.

a) your reasons for your delay in registering this study (after enrolment of participants started);

b) confirmation that all related trials are registered by stating: “The authors confirm that all ongoing and related trials for this drug/intervention are registered”.

Author’s response:

Enrollment in the Korean Clinical Trials Register was delayed after passing the IRB due to documentation errors. We confirmed that one center started before being enrolled in the Korean Clinical Trial Registry. All studies were conducted after passing the IRB, but we acknowledged our mistakes and inserted them into the method based on the reviewer's recommendations..

Please see the page 102-105. 

The trial was registered in the Korean Clinical Trial Registry (KCT 0004595; https://cris.nih.go.kr). Although it was registered with the KCT registry 3 months after the start of the multicenter study due to a paperwork error, the authors confirm that all relevant ongoing trials for this intervention have been registered.

3. Please provide additional details regarding participant consent. In the ethics statement in the Methods and online submission information, please ensure that you have specified whether consent was written or verbal/oral. If consent was verbal/oral, please specify:

a) whether the ethics committee approved the verbal/oral consent procedure,

b) why written consent could not be obtained, and

c) how verbal/oral consent was recorded. If your study included minors, please state whether you obtained consent from parents or guardians in these cases. If the need for consent was waived by the ethics committee, please include this information.”

Author’s response: 

 We have obtained written consent. We specified the contents of the written consent in the main text, and submitted the written consent form at the time of first submission of the manuscript.

Please see manuscript line 93-96. And additional file (consent form for clinical trial subject – English and Korean version)

“Eligible patients who written informed consent were randomly assigned to the 1 L PEG with ascorbic acid group and PICO with magnesium citrate group in a 1:1 ratio using computer-generated randomization”

“This study was funded by TAEJOON PHARM Co., Ltd. Seoul, Korea. The funder had no role in the study design, conduct, analysis, or reporting.”

“This study was funded by TAEJOON PHARM Co., Ltd. Seoul, Korea. The funder had no role in the study design, conduct, analysis, or reporting.”

Author’s response: 

As editor’s recommend, we removed sentence to our manuscripts

Please insert “This study was funded by TAEJOON PHARM Co., Ltd. Seoul, Korea. The funder had no role in the study design, conduct, analysis, or reporting”

Sincerely 

Young-Eun Joo / yejoo@chonnam.ac.kr

Department of Internal Medicine, Chonnam National University Medical School, 160 Baekseo-ro, Dong-gu, Gwangju, 61469, Republic of Korea

---

## [Decision Letter · Decision Letter 1]

12 Dec 2022

Comparison of Optimal Bowel Cleansing Effects of 1L Polyethylene Glycol with Ascorbic Acid versus Sodium Picosulfate with Magnesium Citrate: A Randomized Controlled Study

PONE-D-22-01108R1

Dear Dr. Joo,

We’re pleased to inform you that your manuscript has been judged scientifically suitable for publication and will be formally accepted for publication once it meets all outstanding technical requirements.

Kind regards,

Miquel Vall-llosera Camps

Senior Editor

PLOS ONE

Reviewers' comments:

Reviewer's Responses to Questions

**Comments to the Author**

1. If the authors have adequately addressed your comments raised in a previous round of review and you feel that this manuscript is now acceptable for publication, you may indicate that here to bypass the “Comments to the Author” section, enter your conflict of interest statement in the “Confidential to Editor” section, and submit your "Accept" recommendation.

Reviewer #2: (No Response)

2. Is the manuscript technically sound, and do the data support the conclusions?

Reviewer #2: Yes

3. Has the statistical analysis been performed appropriately and rigorously? 

Reviewer #2: Yes

4. Have the authors made all data underlying the findings in their manuscript fully available?

Reviewer #2: Yes

5. Is the manuscript presented in an intelligible fashion and written in standard English?

Reviewer #2: Yes

6. Review Comments to the Author

Reviewer #2: Minor Revisions: (Page numbers refer to those in the tracked changes version.)

Line 144: Replace "appropriate with target."

7. PLOS authors have the option to publish the peer review history of their article (what does this mean?). If published, this will include your full peer review and any attached files.

Reviewer #2: No

---

## [Editor Report · Acceptance letter]

21 Dec 2022

PONE-D-22-01108R1 

Comparison of optimal bowel cleansing effects of 1L polyethylene glycol with ascorbic acid versus sodium picosulfate with magnesium citrate: A randomized controlled study 

Dear Dr. Joo:

I'm pleased to inform you that your manuscript has been deemed suitable for publication in PLOS ONE. Congratulations! Your manuscript is now with our production department. 

Kind regards, 

on behalf of

Dr. Miquel Vall-llosera Camps 

Staff Editor

PLOS ONE